# Phage-plasmids promote recombination and emergence of phages and plasmids

Eugen Pfeifer [1] & Eduardo P. C. Rocha [1]

Phages and plasmids are regarded as distinct types of mobile genetic elements that drive bacterial evolution by horizontal gene transfer. However, the distinction between both types is blurred by the existence of elements known as prophage-plasmids or phage-plasmids, which transfer horizontally between cells as viruses and vertically within cellular lineages as plasmids. Here, we study gene flow between the three types of elements. We show that the gene repertoire of phage-plasmids overlaps with those of phages and plasmids. By tracking recent recombination events, we find that phage-plasmids exchange genes more frequently with plasmids than with phages, and that direct gene exchange between plasmids and phages is less frequent in comparison. The results suggest that phage-plasmids can mediate gene flow between plasmids and phages, including exchange of mobile element core functions, defense systems, and antibiotic resistance. Moreover, a combination of gene transfer and gene inactivation may result in the conversion of elements. For example, gene loss turns P1-like phage-plasmids into integrative prophages or into plasmids (that are no longer phages). Remarkably, some of the latter have acquired conjugation-related functions to became mobilisable by conjugation. Thus, our work indicates that phage-plasmids can play a key role in the transfer of genes across mobile elements within their hosts, and can act as intermediates in the conversion of one type of element into another.

Mobile genetic elements (MGEs), capable of autonomous transfer between bacterial cells, have a key role in population dynamics and in the evolution of genomes[1,2]. These elements are usually categorized in terms of their mechanisms of transmission. Horizontal transmission can occur when bacterial DNA is packaged into viral particles (phages) or transported between cells through conjugation (conjugative elements). Vertical transmission within lineages occurs either by integration of the element in the chromosome (usually by the means of an integrase)[3,4] or by its establishment as an independent extra-chromosomal replicon (plasmid)[5]. Traditionally, temperate phages are thought as elements integrating chromosomes and conjugative elements as extra-chromosomal elements (plasmids), even if integrative conjugative elements are also very frequent[6]. These different MGEs co-exist in the same genomes where they may interact and exchange genes[7,8]. The fact that plasmids and integrative phages have different mechanisms of horizontal and vertical transmission, means they often lack homologs with high sequence similarity for homologous recombination. Yet, some elements have been described as phage-plasmids[9] or prophage-plasmids[10] (P-Ps): elements that transfer horizontally between cells as phages and vertically within cellular lineages as plasmids[9,11]. These elements encode viral particles and packaging machinery homologous to those of phages and replication initiators and partition systems homologous to those of plasmids. A few of these P-Ps have been known for decades and studied in detail[9,12], but most have never been studied. Recent genome analyses suggests that P-Ps are numerous, accounting for around 5–7% of all plasmids and of all phages[13]. As a point of comparison, only 23% of the plasmids are thought to be conjugative[14]. P-Ps are particularly interesting in that

[1]Institut Pasteur, Université Paris Cité, CNRS UMR3525, Microbial Evolutionary Genomics, 75015 Paris, France. ✉e-mail: eugen.pfeifer@inrae.fr; erocha@pasteur.fr

they have many homologs with other phages and with other plasmids. As a result, one might think that they could exchange genetic information with other types of phages and other types of plasmids, thereby connecting the two worlds of MGEs that are usually regarded as evolving independently.

Our previous work showed that the majority of P-Ps are grouped in a few large families of divergent elements, which suggests that such groups are ancient[13]. Hence, if there are highly similar homologs between P-Ps and other types of elements this suggests recent gene flow between them or interconversions between elements, e.g., P-Ps mutated to become other types of elements. In the first case, one expects to find a few very similar genes between two dissimilar elements. In the second case, one expects to find many very similar homologs between the two elements. Several reports have described plasmids very similar to P-Ps that have lost the capacity to transfer as phages[15–18], i.e. they are now just plasmids. In theory, P-Ps could also generate novel phages that are not plasmids, although that has not been described (to the best of our knowledge). How such transitions occur, and what happens to them, is not well known.

Recombination between MGEs can occur by different mechanisms. If the elements have highly similar regions, then exchanges can occur by homologous recombination. The effect is heightened by the presence of genes encoding recombinases that are more tolerant to sequence divergence in these elements[19,20]. Some MGEs encode proteins facilitating non-homologous end joining which may result in exchanges of different sequences[21]. Finally, transposable elements and integrons may lead to the translocation or copy of regions of one element into another[22–24]. This gene flow may come with a price. The new genes may not be adaptive, and even be costly, recombination may split adaptive combination of genes (or create maladaptive ones) and integration of transposases may lead to gene inactivation. This may affect the ability of the element to transfer and result in its degradation. Genetic exchanges can also be adaptive, i.e. increase the fitness of the element, either by enhancing its ability to transfer horizontally or vertically. For example, variations in the tail fibers of phages may change their host range[25], acquisition of defense systems may protect them (and the host) from other elements (such as virulent phages)[26], and acquisition of anti-defense systems may protect MGEs[27]. Finally, some traits that are unrelated to the biology of the MGEs, such as virulence factors and antibiotic resistance, may allow them to confer advantages to their hosts[28,29], which will favor the increase in frequency of the element (along with that of the host).

Some studies in the last few years identified highly similar homologs in phages and plasmids, including defense systems[30], virulence factors[31], and more recently antibiotic resistance[32]. This raises questions on the frequency and type of exchanges between different types of elements. Here, we analyzed a large number of phages, plasmids and P-Ps to understand if there is gene flow between them. To tackle this question, one cannot use classical phylogenetic methods, because these elements evolve too fast and recombine too much to allow the inference of meaningful deep phylogenies. We thus turned to simpler, and conservative methods: we searched for almost identical genes within very different elements. This allowed to quantify relatively recent gene exchanges (because the sequences are yet very similar) and to test if they are mediated by P-Ps. When searching for homology between P-P and the other elements that could be markers of gene flow, we realized that several plasmids and phages had many homologs with P1-like P-Ps. This revealed that some elements are derived from P-Ps by gene loss accompanied with the acquisition of novel functions. Hence, P-Ps favor gene flow between phages and plasmids and are sometimes the source of novel plasmids and phages. This led us to further investigate three aspects of gene flow between these elements: the functions that are exchanged, the recombination processes that may facilitate these exchanges, and how they spur the conversion of one type of element in another. For simplicity, and

unless otherwise stated, in this work we will distinguish P-Ps on one side and the remaining phages and plasmids on the other, although it should be emphasized that P-Ps are also both phages and plasmids.

## Results

### Phage-plasmid gene repertoire overlaps with those of phages and plasmids

We started our analysis by establishing a network of sequence similarity between all phages, plasmids and P-Ps. First, we classed the MGEs in types. We grouped each type of element with the standard tools for that element to facilitate linking these results with the available literature. Most (2412 out of 3585) phages were clustered into 258 viral clusters (VCs)[33]. Out of 20,274 plasmids, we could group 9383 into 356 plasmid taxonomic units (pTUs)[34]. Finally, we used our previous classification[13] of 1146 out of 1416 P-Ps in 9 groups (Supplementary Dataset S1). Unassigned elements were classed as singletons. Of note, these elements were identified computationally using a method that searches in plasmids for the set of essential phage functions (and vice versa in phages). They should thus be bona fide P-Ps, although we cannot ascertain that all these elements are fully functional. Even if some elements may be unable to develop a full infectious cycle, they are still in genomes and available for recombination exchanges. We then assessed the similarity among groups using gene repertoire relatedness (wGRR), an index accounting for the fraction of genes of the smaller element that have a bi-directional best hit in the largest element. The index is weighted by the sequence similarity of the homologs (see Methods). We computed the wGRR between each pair of elements and then made a mean per group (Supplementary Dataset S2). This allowed to build a graph where nodes are groups (singletons were excluded for clarity) and edges represent the average wGRR values between the groups (Fig. 1A). This revealed a highly connected graph where most groups have some homologs in other groups.

The graph allows to assess the network of relations of similarity between elements of the same type and between types. Since the mesh of nodes and edges is hard to scrutinize, we arranged the nodes using a force-based algorithm (Fruchterman-Reingold)[35]. This method uses the weight of the edges to attract nodes and the absence of edges between nodes to separate them. This representation reveals plasmids on one side of the graph and phages at the opposite position, while most P-P groups are in an intermediate position between the two because they have homologs to both types of elements. A few P-Ps, such as those belonging to the group of CampHawk (mostly from *Bacillus*) or Actinophages_A (mostly from *Mycobacteria*) were closer to phages and others, like the cp32-like P-Ps (from *Borrelia*)[36], closer to plasmids. Overall, the arrangement indicates a continuum of MGEs with P-P groups placed between phages and plasmids.

### Phage-plasmids drive gene exchanges between phages and plasmids

The presence of homologous genes in different MGEs can result from either common ancestry (vertical transmission) or genetic exchanges. Ancient events of genetic exchange are challenging to detect since their tracks may be blurred by mutations or re-occurring recombination events. Furthermore, phages and plasmids evolve rapidly by recombination and their ancestral lineages cannot be obtained (because different genes have different histories). Thus, we focused our analysis on the detection of gene exchanges at the short time scale. Since we cannot use phylogenies to trace events of exchanges, we used a previously published approach aimed at identifying highly similar genes in very different elements[37]. Briefly, we define that one pair of homologous genes participated in a recent gene exchange event if they are very similar (>80% identity, >80% alignment covering sequence) and are in very different MGEs (wGRR <0.1). We excluded events between pairs of MGEs with more than 25 exchanges from further analysis, because these were associated with very large

replicons that may be chromids or secondary chromosomes (see "Methods"). All other pairs of MGEs were analyzed, i.e. the analysis is not restricted to comparisons between groups of elements (as in Fig. 1). Genes identified as being part of exchanges were named recombining genes (RGs) and the others were named non-recombining genes (NRGs) (Fig. 2A). The distribution of sequence similarity among RG and the remaining genes shows the former are indeed too similar to result from high sequence conservation since the last common ancestor of both MGEs (Fig. S1). The fraction of genes classed as RGs varied widely with the type of MGE: 4.7% of phage genes, 14.9% of P-Ps, and 27.1% of plasmids. Hence, plasmids seem to exchange more and phages less, with P-Ps having an intermediate rank (Fig. 2B, Fig. S2).

The fraction of each element gene repertoire that is marked as RG varies a lot. Many plasmid genomes have more than 50% of the genes classed as RGs. One should note that each genetic exchange detected by our method identified only small parts of the elements (since we constrain the search to pairs of elements with wGRR<0.1). Hence, it automatically excludes cases of comparisons between P-Ps that might result from co-integration of phages and plasmids and other related phages and plasmids. Thus, the identification of a high frequency of RG in an element means that it has exchanged different genes with different elements. The frequency of RGs in P-Ps differs between groups. On average, P1-like P-Ps have many RGs (ca. 30% of each element)

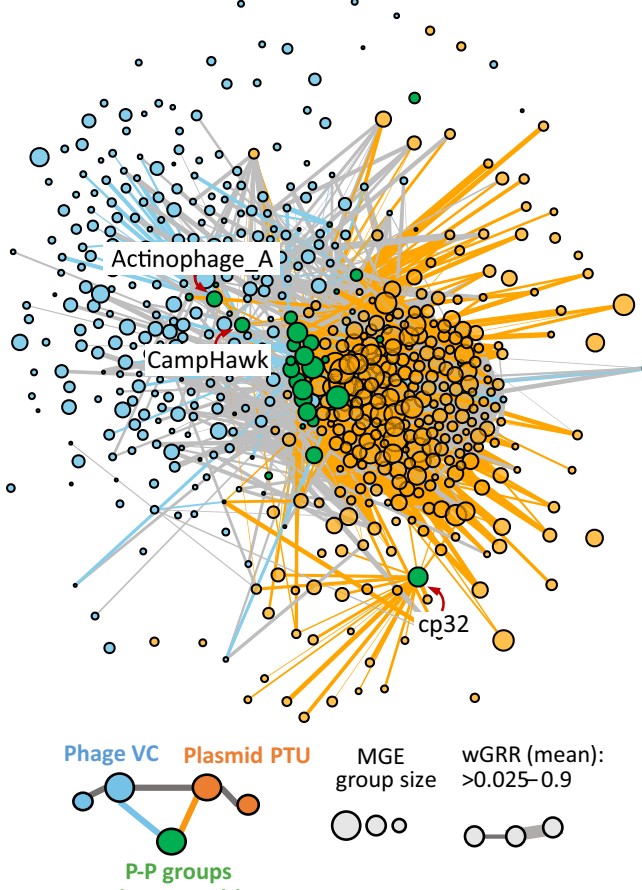

**Phage VC** **Plasmid PTU** MGE group size wGRR (mean): >0.025– 0.9

**P-P groups and communities**

**Fig. 1 | The network of weighted gene repertoire relatedness (wGRR) between groups of phages, plasmids and phage-plasmids (P-Ps).** The nodes represent groups of elements and the edges connect groups with a mean wGRR larger than 0.025 (and are weighted by the mean wGRR). Edges linking phage and P-P are in blue, those between plasmid and P-P are in orange and those between phages or between plasmids are in gray. The node arrangement was produced by using a force-directed (Fruchterman-Reingold) algorithm[35].

whereas N15-like P-Ps have fewer (ca. 20%) (Fig. S3). This fits our previous observation that P1-like genomes are more diverse in terms of gene repertoires than other P-Ps (such as members of the N15 group)[13]. Of note, we could not find any homologs for some genes, which were thus classed as NRG-nh. These were 6.1% of plasmids, 7.8% of P-Ps and 11.0% of phage genes (Fig. S2, Supplementary Dataset S3). In conclusion, our analysis confirms that many genes in MGEs were subject to recent genetic exchanges.

We then tracked the exchanges between all types of MGEs (Fig. S4). The assessment of these exchanges is complicated by the inability to identify the direction and origin of genetic exchanges in the absence of a phylogenetic tree regrouping all the elements. To accommodate this, we grouped RGs in gene families (see Methods). We obtained 44,634 families of RGs from phages, plasmids and P-Ps (Supplementary Dataset S4). We then assessed the pairs of events per gene family within and between MGE types (Fig. 2C, see sketch of the analysis in Fig. S5). More precisely, we tested if within a family of RGs there were exchanges within plasmids, P-Ps and phages, and between these different elements. Each type of exchange was only counted once to lower the probability of counting multiple times the same ancestral exchange event (see "Methods"). Most RG families were detected in plasmids, which is around ten times the number of RG found in phages and P-Ps. Accordingly, there are many more exchanges between plasmids (93.5%) than between phages or P-Ps. This reflects the abovementioned high frequency of RGs in plasmids, but also the fact that the set of plasmids includes many more genes. Interestingly, despite P-Ps being the least abundant of the MGEs they exchange many more genes with both plasmids and phages than the exchanges observed between phages and plasmids. To compare these values with random expectations we made simulations where we took the labels of the elements (P-P, plasmid, phage) and randomly attributed them to the MGEs (this keeps the proportions of each element). The comparison of random expectation and observed values shows that among exchanges between different types, those between P-Ps and phages or plasmids are much frequent than expected by chance (respectively 1.26 and 2.37 times more frequent), whereas those between phages and plasmids were rarer (only 16.7% of the expected ones). Only 11.2% of the exchanged gene families encode typical plasmid (replication, partition) and phage functions (lysis, tails, connector, head and packaging), showing that the panel of functions that are exchanged is very diverse. P-Ps even exchange more with plasmids (2.3 times) than with other P-Ps. Even if our procedure is very conservative and we ignore most MGEs, a few RG families are found in all types of MGEs (Supplementary Dataset S4). Taken together, P-Ps have more recombination events with plasmids or phages, than phages and plasmids directly. This suggests that gene flow between plasmids and phages may take place via exchanges with P-Ps.

## Functions exchanged by phage-plasmids with plasmids and phages

We aimed to identify the functions transferred between different mobile genetic elements. Using specialized databases (see Methods, Supplementary Dataset S3), we could annotate 38.4% of RGs and 21.9% of the NRGs (Fig. S6B). For each MGE type, we compared the frequency of RGs and NRGs in each functional group and tested if they are significantly different (Fig. 3, Supplementary Dataset S5). In addition, we compared the number of RGs exchanged within and between types of MGEs (Fig. S5B–F, Supplementary Dataset S6). We focused initially on core phage and plasmid functions and then in three groups of accessory functions.

We used PHROGs, a database of profiles of phage functions[38], to annotate the phage-associated functions in P-Ps (39.8% of genes) and phages (41.5% of genes) and could even annotate a few in plasmids (14.5% of genes) (Supplementary Dataset S5). Genes encoding tail and head structures, packaging, lysis, and functions involved in

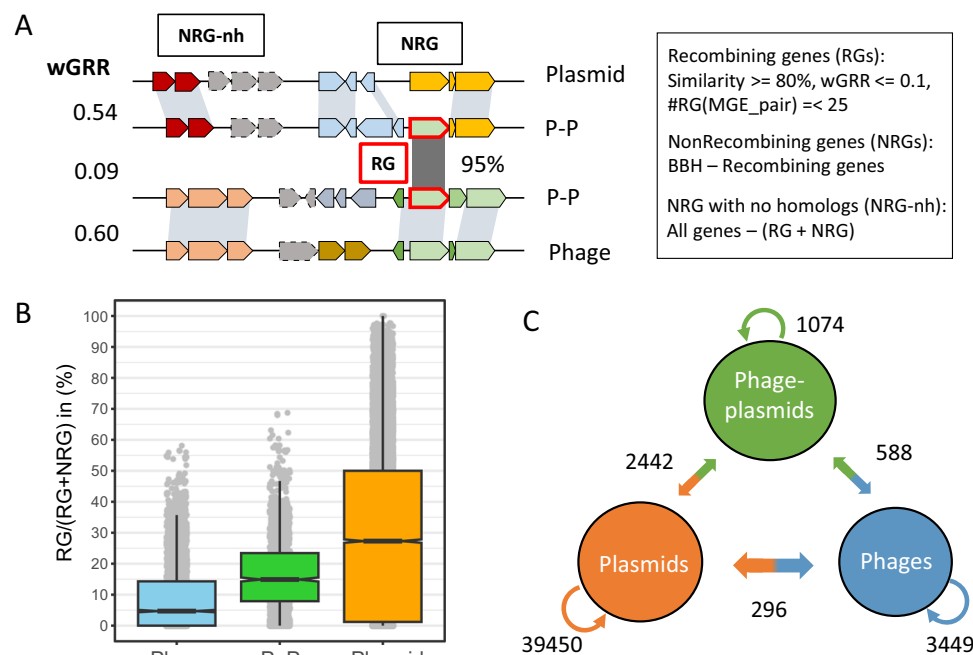

**Fig. 2 | Gene flow between phages, plasmids and phage-plasmids. A** Allocation of genes from MGEs into recombining (RG), non-recombining genes (NRG) and NRGs with no homologs (NRG-nh). **B** The box-plot represents the frequency of RG (relative to the sum RG plus NRG) in MGEs (n(P-P) = 1416, n(Phage)= 3575 and n(Plasmid)=20,057). In the box plot (notch style), the whiskers mark the 1.5 inter quantile length from the 1st (box bottom line) to the 3rd (box top line) quantile. Of note, a few phages and plasmids do not have RGs or NRGs (but only NRG-nh), and were excluded. **C** The numbers on the arrows represent the number of RG families with at least one such type of exchange (e.g., at least one exchange between plasmids in orange or between phages and plasmids on the orange to blue arrow).

transcriptional regulation are significantly overrepresented in RGs in phages (ca. 28% of RGs), but not in P-Ps (14.2% of RGs of P-Ps) nor in plasmids (<0.7% of RGs). Genes encoding heads and packaging functions are enriched in phage-to-phage or P-P-to-P-P transfers, but not between the two groups, suggesting some sort of specificity of these functions (Fig. S6C). The few phage functions encoded by plasmid RGs were overrepresented in genes that are transferred among different types of MGEs, indicating that they were likely acquired from P-Ps or phages. We also searched for plasmid core functions. Replication and partition systems are significantly more frequent among RGs and especially among the ones exchanged between MGE types (here plasmids and P-Ps, although enrichments are low with $R_{Diff-Sum} < 0.13$) (Fig. S6D, Supplementary Dataset S6).

Anti-MGE defense systems are often encoded in MGEs[39] and are frequently gained and lost[26,40], suggesting that their repertoire depends on gene flow between MGEs. We searched for complete defense systems and smaller variants of these (because MGEs often have shortened versions of known defense systems[41], see Methods). They represent a notable fraction of RGs: 2.6% in phages, 8.5% in P-Ps, and 6.2% in plasmids (Supplementary Dataset S5, Fig. 4). Examples of full systems are shown in Fig. S7 and smaller variants in S8. Defense genes are substantially enriched in RGs (Fig. 3B). Moreover, defense genes identified by an approach with higher profile coverage ("Defense-05") are significantly overrepresented in genes that are transferred between different MGE types (Fig. S6E).

Antibiotic resistance genes (ARGs) are common in plasmids, frequent in P-Ps and rare in phages[29,42]. ARGs in plasmids and P-Ps are often identical suggesting that they were recently exchanged. A few ARGs had highly similar homologs across phages, plasmids and P-Ps, one case consisting of a gene family encoding resistance against chloramphenicol (Fig. S9). ARGs are overrepresented in RGs and they are often found in inter-MGE type exchanges, most frequently between plasmids and P-Ps (excess of $R_{Diff-Sum} = +0.50$) (Fig. S6D, Supplementary Dataset S6). This fits our previous observations that P1-like

elements acquired ARGs by exchanges with plasmids mediated by IS-elements, integrons, and other recombinases[32].

Known toxins and virulence factors (VFs) are a small fraction of the genes in MGEs (0.05% of phage, 0.23% of P-P and 1.4% of plasmid genes). Some VFs, such as *gtrB*, a glucosyl transferase involved in the modification of the O-antigen[43], are detected in phages, plasmids, and P-Ps (Fig. S9). Only in phages, VFs are overrepresented in RGs, and in genes that are transferred between different types of MGEs (Supplementary Dataset S6). Nearly half of the phage VFs in the RG group (48%) are Shiga toxins (*stx* genes), which are found in highly diverse temperate phages (known as stx-phages). Stx-phages can infect a wide range of different hosts and have a crucial role in the pathogenicity of Shiga toxin-producing *Escherichia coli* strains. It was suggested that the spread of *stx* genes is driven by stx-phages[44]. Our analysis indicates that phages are also involved in the spread of virulence factors through transfers to other MGE types.

## Contribution of transposases and recombinases to gene exchanges

What are the mechanisms driving the genetic exchanges between so widely different mobile genetic elements? Such exchanges could be driven by recombinases that mediate site-specific or homologous recombination or by transposases that mediate translocations. Phages and some other MGEs are known to encode recombinases that require shorter regions of homology and lower sequence similarity for recombination and may thus facilitate exchanges between distant elements[45]. We annotated these functions to assess their contribution to gene exchanges (see Methods). The tests for enrichment and depletion showed that recombinases and transposases are strongly overrepresented in RGs (Fig. 4B). Similar results were obtained by the analysis of the PHROG category "integration and excision", which also showed an excess of these genes in RGs in all MGE types and among the genes exchanging between MGE types (Fig. 4 and Fig. S11). For example, 14.7% of RGs of plasmids, 15.2% of

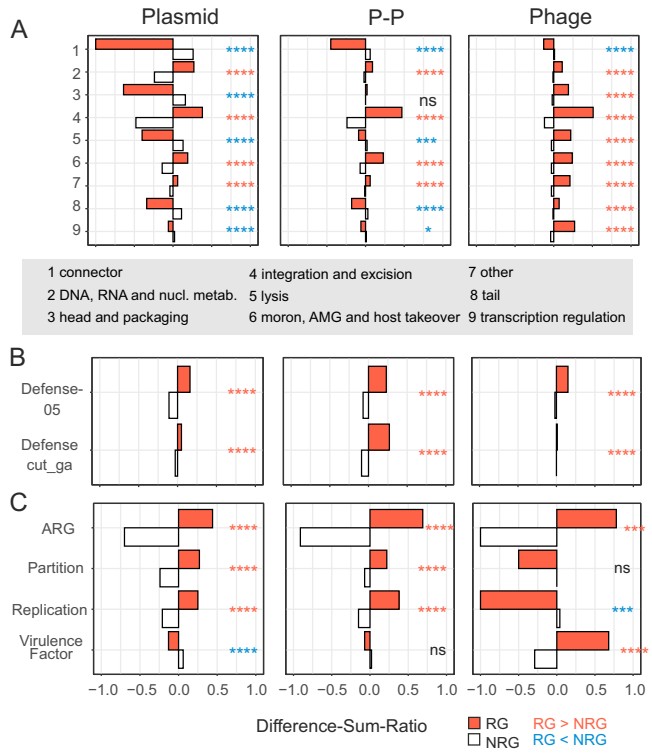

**Fig. 3 | Functions enriched or depleted among recombining genes (RGs). A–C** RGs were annotated using different databases (see "Methods", auxiliary metabolic genes (AMGs)). The numbers of RGs per category and type of mobile elements were compared to the number of non-RGs (NRGs) using a one-sided Fischer's exact test with the Benjamini-Hochberg correction. The tests were two-tailed and when RGs were significantly different from NRGs this was marked by red stars if their frequency was larger and by blue stars if their frequency was smaller than that of NRG. Panels are organized in the way in which the groups were tested (to guarantee independency). In particular, genes assigned to multiple categories were not tested together, e.g. genes annotated by Defense and DNA metabolism (of PHROGs). Genes with no matches to any profile or database are not shown but were included in the tests and their numbers are listed in Supplementary Dataset S5, including all *p* values of all conducted tests. Difference-sum ratios were computed (see "Methods") as the ratio of the difference between observed and expected over the sum. pvalue: ns >0.05, * <=0.05, ** <= 0.01, *** <=0.001, **** <= 0.0001. **A** Genes were annotated by PHROG profiles. Numbers in the y-axis represent categories that are listed in the gray box. **B** Gene annotation was done by profiles taken from DefenseFinder[79] (see "Methods"). **C** AMRFinderPlus[80], VFDB[83] and profiles matching plasmid partition and replication functions were used to annotate RGs and NRGs.

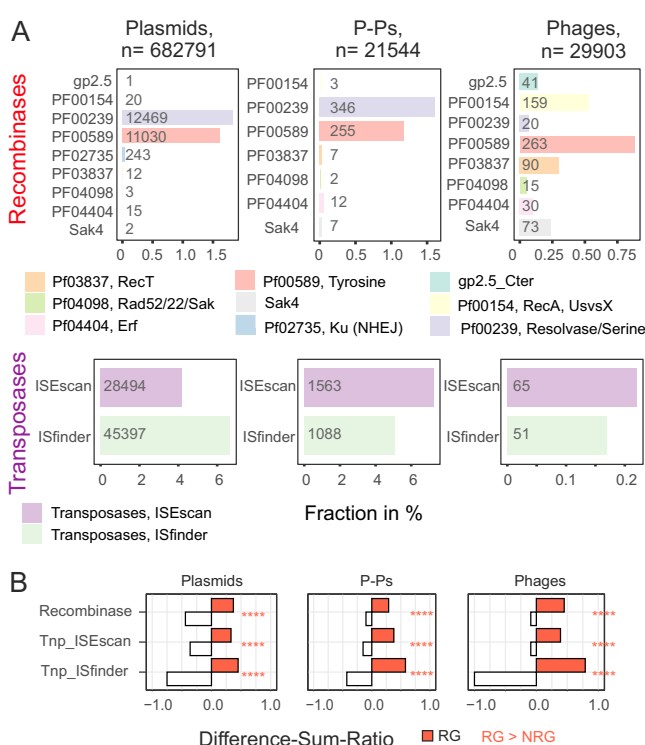

**Fig. 4 | Recombinases and transposases in plasmids, phage-plasmids and phages. A** Recombinases and transposases were detected in all recombining genes (RGs) (see Methods). Shown is the fraction (in %) of RGs in all types of mobile elements (counts are shown in the columns). **B** Enrichment analysis were done as shown in Fig. 3 for recombinases and transposases in RGs vs non-RGs.

Overall, transposases and recombinases represent a large fraction of RGs, especially in plasmids and P-Ps. They promote genetic exchanges within MGE types and facilitate exchanges between different types of MGEs.

## Many plasmids and a few integrative prophages are closely related to P1-like phage-plasmids

The analysis of homologies between plasmids, phages and P-Ps revealed some very similar elements of different type. These intriguing similarities could provide insights into either the genesis of P-Ps or their fate (and how they are shaped by genetic exchanges). We searched for homologous sequences of P-Ps in 3585 phages, 20274 plasmids and 50262 integrated putative prophages. We focused on 149 P-Ps of the P1 group and searched for similar plasmids and phages (wGRR >0.1 and a minimum of 10 homologs, Fig. 5A). We detected 45 plasmids and 12 integrated prophage regions, most of which were assigned to P1-like P-Ps of the subgroup 1 (P1g1, where P1 is included, examples pointed out in Fig. S12). We analysed these elements in the light of the conserved gene repertoires of P1-like P-Ps[32] (Fig. S13). Some plasmids and integrative prophages have an almost complete set of such genes whereas others are more distantly related and only have a few homologs (Supplementary Dataset S7).

We then focused on the elements with at least 75% of the genes that are conserved in the P1g1 group. This included 3 integrative prophages and 38 plasmids. We pooled these elements together with 119 P1g1-like P-Ps and computed a pangenome of the set using strict thresholds (80% identity of an alignment covering 80% of each sequence) (see Methods). We found 17 gene families conserved in at least 90% of these elements (Fig. 5B), often in a few conserved genetic neighborhoods (Fig. 5C) suggesting that they are of functional importance. These families included several phage functions (tails, head, packaging, lysis). Some functions often associated with plasmids

P-Ps and 2.7% of phages were found to encode these proteins (Fig. 4A). This is much more than their frequency among NRGs (<2% in plasmids, <4% in P-Ps, and <1% in phages) (Supplementary Dataset S5). Among these genes, transposases are the most abundant and have very specific patterns of distribution. They are more abundant than recombinases in plasmids and P-Ps (11-12% of the plasmid RGs) but very rare (<0.4%) in phages. Recombinases are less frequent overall, but more frequent than transposases in phages (2.3% of the phage RGs). Typical phage recombinases like Sak4 and RecT are relatively rare in P-Ps.

Transposases promote their own transfer and may co-transfer neighboring genes (e.g. composite transposons). We inspected RGs and the genes adjacent to them to class them in: single RGs, genes flanking RGs, and RGs adjacent to other RGs (Fig. S11A). Most transposases are single RGs (such as IS elements) or RGs adjacent to other RGs. They are found in similar frequencies in plasmids (ca. 10%) and P-Ps (10–15%), but very rarely in phages (0.3–0.5%). We found that few NRGs flanking RGs are transposases (ca. 3% in plasmids and P-Ps, and <1% in phages) (Fig. S11D).

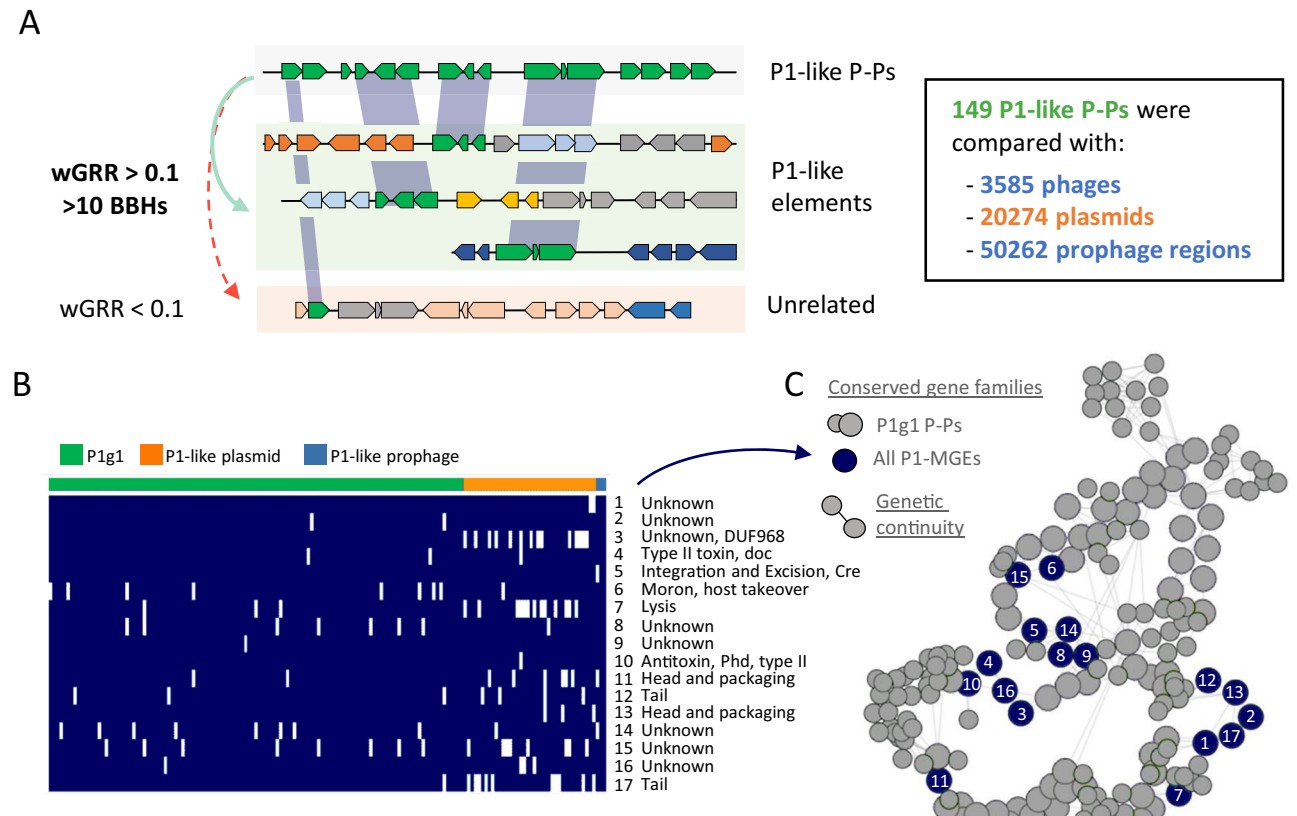

**Fig. 5 | P1-like MGEs have 17 gene families in common. A** The screen for homologous P1-like elements resulted in 45 plasmids and 12 integrative prophage regions, of which some are closely and others are weakly related. **B** Closely related elements (having at least 75% of conserved gene families of the P1 subgroup1) were grouped (119 P1g1 P-Ps, 38 plasmids and 2 integrative prophage regions) and 17 persistent genes (present in more than 90% of the elements) were identified using PanACoTA (see "Methods"). **C** Pangenome graph of the P1 subgroup1 (P1g1)[32]. The conserved gene families of the P1-like elements (from **B**) are labeled as dark blue nodes. Gray nodes correspond to other conserved gene families (large nodes with high and small with intermediate frequency in all genomes) in the pangenome of P1g1. Edges represent the conservation of the genetic neighborhood. If an edge is assigned between two nodes, then the respective genes are co-localized.

were also present, including the *phd/doc* addiction module (toxin-antitoxin system)[46] and the Cre recombinase that is important for the multimer-resolution of P1[47]. Interestingly, we could not find a partition system and a replicase among the conserved genes, which suggests these were lost or exchanged, possibly to avoid incompatibility with extant P-Ps. Of note, although the 38 plasmids have many homologs in P1-like elements, we could not identify them with our random forest models designed to detect confident P-Ps[13]. This is because many of these plasmids are depleted in key phage functions (which are required to detect P-Ps). For instance, 7/38 of the plasmids lack lysis genes (P1 has seven genes matching the lysis category), 9/38 have only two different tail genes (in contrast to 15 different genes in P1), and 7/38 lack any transcriptional regulator (needed to regulate the lysogenic or lytic cycles) (Supplementary Dataset S7). Overall, the absence of these genes indicates that these plasmids cannot undergo a full phage life cycle.

In conclusion, P1-like P-Ps have closely related plasmids and integrated prophages, the former showing signs of being defective P-Ps. That P-P-like elements can be detected as (chromosomal-) integrated elements highlights how some P-Ps may be propagated in bacterial chromosomes.

### Plasmids and integrative prophages derived from P1-like phage-plasmids
To understand the evolutionary relation between P1-like P-Ps and the other elements, we built two rooted phylogenetic trees (mid-point rooted and outgroup rooted) of the 17 conserved gene families (Fig. 6,

Fig. S14). The three integrative prophages were closely and confidently placed in two branches next to P-Ps. This suggests an order of events where these elements derived recently from P-Ps that became integrative prophages. These elements encode replication and partition genes similar to those of P1. The cause of these integration events remains to be clarified. An initial analysis of the genes flanking the regions homologous to P1-like P-Ps showed that the neighboring genes encode recombinases or are assigned to prophage genes (examples pointed out in Fig. S12). This indicates that P1-like P-Ps integrate regions in proximity to resident integrative prophages.

Most of the P1-like plasmids (30/38) were grouped together in a single, well-supported branch in both rooted trees showing that plasmids are derived states in the tree, i.e., the order of events is that P-Ps became plasmids and not the other way around. There are relevant genetic differences between P-Ps and these plasmids, e.g. 18 plasmids are classed into a specific taxonomic unit (pTU-E49), whereas the rest could either not be classed or were classed with P-Ps in the pTU-Y. We typed all the elements in terms of replication compatibility (Inc-typing, see Methods) and found that most plasmids and P-Ps are classed in the groups p0111 (28/38 plasmids, 63/119 P-Ps) and IncY (8/38 plasmids, 52/119 P-Ps). In addition, whereas 50% of the plasmids were classed in one group (either as IncY or p0111), the other half was classed in several other Inc-types, suggesting that they have become compatible with other P1-like plasmids and P-Ps in the same host.

The analysis of the gene repertoires suggests that there were gene exchanges concomitant with the transition of P-Ps into plasmids. In most of the plasmids (N = 30), we detected either a relaxase gene

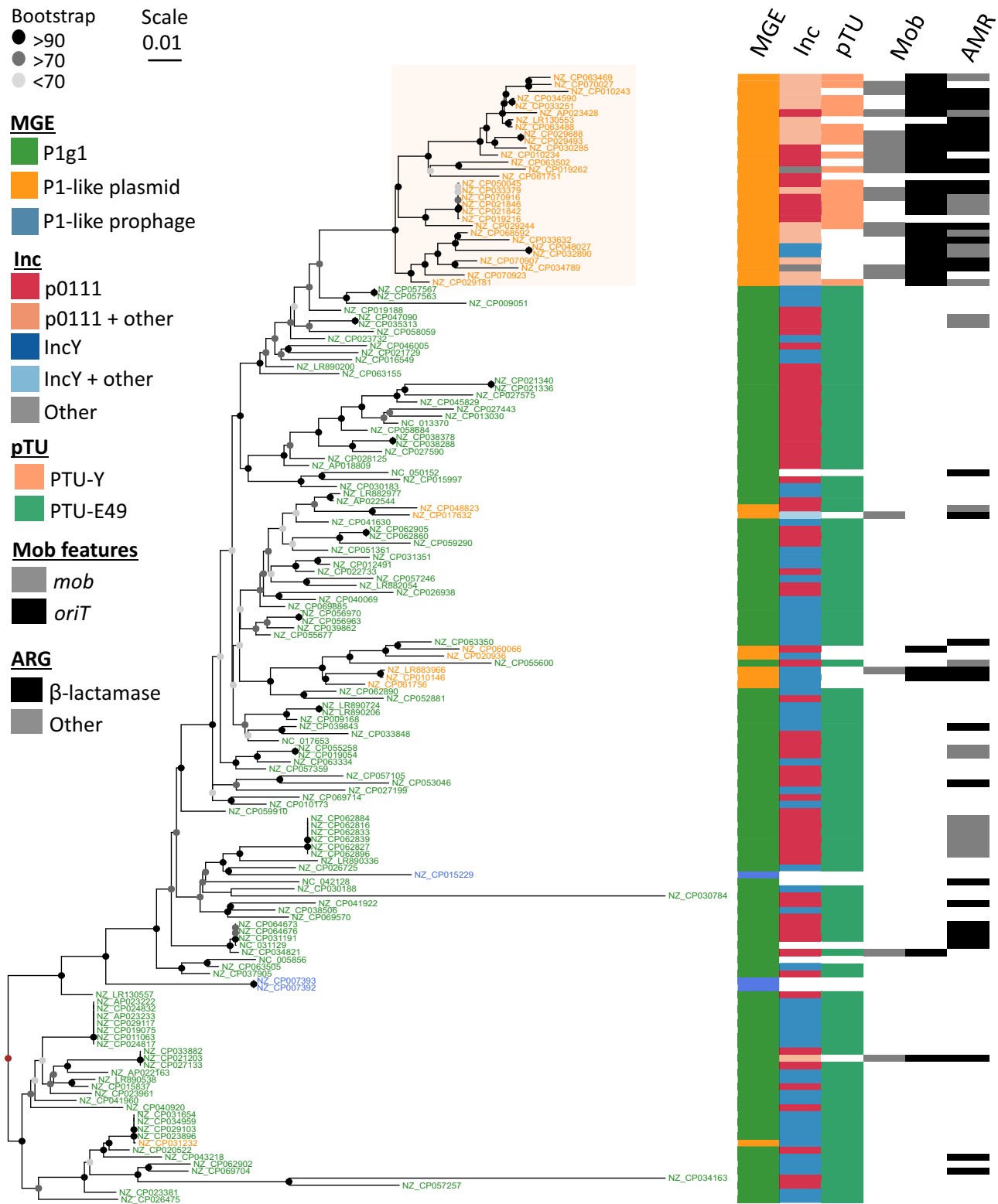

**Fig. 6 | Phylogenetic tree of P1-like mobile elements.** A global alignment of the 17 persistent gene families was done and given to IQ-TREE v2 (see "Methods") to compute a phylogenetic tree by maximum likelihood with 1000 ultrafast bootstraps (best model: SYM + I + G4). Visualization was done with ggtree[89] in the R environment. Origins of transfers and relaxases were searched in plasmids and phage-plasmids in a previous work[48]. The tree was rooted using the method of the midpoint-root (brown node) and confidence of the bootstrap values are indicated by the node colors ranging from light gray (less than 70%, uncertain branches) over gray (>70–90%, moderate) to black (>90%, confident branches). The NCBI accessions IDs of the elements are indicated for plasmids and P-Ps in the terminal branches of the tree. For integrative prophages, the ID of the bacterial chromosome is shown with details on the prophage given in Supplementary Dataset S7.

(*mob*) or an origin of transfer (*oriT*). These genetic elements are absent from all but two P1-like P-Ps. The *mob* gene and the *oriT* are needed for conjugative transfer of plasmids, suggesting that many plasmids derived from P1-like P-Ps have lost the ability to transfer horizontally as phages, but have recovered genetic information allowing them to be mobilized by conjugative elements[48]. Interestingly, while ARGs are found in many P1-like P-Ps, they are even more frequent in these novel plasmids, where 29/38 plasmids encode ARGs, especially against β-lactam antibiotics.

Taken together, these results show that P-Ps are a genetic source of plasmids and integrative prophages and that this occurs by evolutionary processes involving the exchange and loss of genes.

## Discussion

Genetic exchanges within one type of MGE have been widely studied, e.g. between plasmids[49–51] or between phages[8,37]. However, very few studies searched for gene exchanges between these different types of elements. Here, we found many and we believe they are only a fraction of all exchanges, because our analysis is conservative. First, we are very strict in the definition of RGs, such that we require high sequence identity between the genes and very little similarity across the rest of the elements. By this approach we found nearly all gene pairs (92.5% between recombining elements as shown in Fig. S1) to be either not similar or classed as RGs, meaning weakly related genes are rarely present. This supports the view that genes identified as RGs were subject to recent exchanges instead of being the product of strictly vertical transmission. There are certainly many exchanges between more similar MGEs that we missed. Yet, since we focus on exchanges between very different types of elements, our analysis may have captured many of the most interesting events. Second, we count only types of events (e.g., exchange of a gene family between P-Ps and plasmids) even if they may represent multiple independent events of exchange. Finally, it should be noted that when we identify an exchange this may represent one single exchange between ancestors of these elements or it may represent a chain of successive genetic exchanges between multiple MGEs that included the ancestors of the two focal elements.

We identified very frequent gene flow within plasmids, less frequent within phages, and intermediate between P-Ps. When we analyzed exchanges between types of elements we found that they tend to occur with P-Ps (both with phages and with plasmids), even if these elements are the least represented in our dataset. Why are P-Ps more prone to participate in such gene flow? Contrary to the plasmids and integrative phages that have few homologs, these elements systematically have homologous genes with both phages and plasmids and this may facilitate exchanges by homologous recombination. The existence of homologs to integrative phages and plasmids that are important to the functioning of P-Ps may also facilitate the fixation of such exchanges, since phage and plasmid-associated functions may both be adaptive in P-Ps. Yet, it should be emphasized that our functional analysis showed that core plasmid or phage functions were a minority of the RGs. Another trait of P-Ps that may explain their high frequency of RGs is that they tend to be larger than the average integrative phages[13]. This makes them more tolerant to accretions of genetic material, which can block packaging of phage DNA in the viral capsid. This is because the efficiency of packaging in phages decreases rapidly with genome extensions[52]. Hence, acquisitions of many genes in one single event of exchange may result in an element that cannot develop a complete lytic cycle because it's is incapable of packaging its genome in its own capsid. Finally, P-Ps have extensive regions of accessory plasmid-like regions that may be more tolerant to interruption than the integrative temperate phages. The latter tend be organized in a few long operons with essential genes complicating the integration of novel material without deleterious consequences for gene expression[53,54]. The hybrid nature of P-Ps may

thus contribute to their high rate of genetic exchanges with both phages and plasmids.

What are the mechanisms driving these gene exchanges? Some mechanisms are nearly impossible to determine because they leave little evidence in the genome. This is the case for events taking place by homologous recombination or by NHEJ[37]. Other mechanisms can be identified by sequence analysis, notably transposable elements and phage recombinases. We found many transposable elements jumping between elements and especially between P-Ps and plasmids that have recently exchanged genes. Transposases are often RGs or part of a set of adjacent RGs. Given space constrains in the genomes of phages and P-Ps, non-adaptive parts of transposable elements may be lost quickly after transposition. When this happens, the comparisons between elements will fail to reveal the presence of transposases and one may spuriously infer an exchange without ISs. Hence, to identify the mechanisms of exchange it is crucial to analyze recent events. The important role of transposable elements driving intragenomic genetic mobility for gene exchanges between P-Ps and plasmids fits our previous work showing that antibiotic resistance genes that have recently been acquired by P-Ps[32] tend to be found in integrons and/or be flanked by transposable elements. It has been extensively described that transposable elements are key drivers of exchanges between plasmids[24,55,56]. Our results suggest that ISs also have an important role in gene exchanges between P-Ps and plasmids.

Many diverse functions are exchanged between P-Ps and the other elements. We found frequent exchanges of phage-related functions within phages and within P-Ps, but less frequently between the two groups. P-Ps, like the other phages, show specificity in the exchange of some components of the viral particle (i.e., they exchange more within P-Ps than with phage). In contrast, plasmid functions such as replication and partition genes are very rare in phages, frequent in P-Ps (and plasmids) and were often transferred between plasmids and P-Ps. These genes are required for stability and maintenance of episomal elements, and very similar systems cause errors in the propagation of the element, ultimately creating incompatibility between plasmids and resulting in their loss[57]. The observed frequent transfer of these systems may allow P-Ps and plasmids to escape the incompatibility with other elements present in the same cell. Among accessory functions exchanged between different types of MGEs, we found many defense systems, and especially restriction-modification systems (Supplementary Dataset S3). Some of these elements not only render MGEs more competitive with other MGEs but may also allow them to spread selfishly because they are poison-antidote systems (e.g., Restriction-Modification systems)[58]. The observed frequent transfer of antibiotic resistance[32] and virulence factors[59,60] matched previous observations. Of importance is the fact that we could not identify a function for most exchanged genes. Many interesting unknown adaptive traits may yet remain to be found among these genes.

Gene gains and losses can result in the inactivation of MGEs. Defective prophages are numerous in bacterial genomes[61,62] and plasmids with defective conjugative systems have also been described[14]. P-Ps are somewhat special in that gene loss might result in elements that are just plasmids or just integrative prophages. Here, we found that gene loss and genetic exchanges in P1-like P-Ps resulted in the generation of such MGEs. As mentioned above, our analysis cannot ascertain if the integrated P-Ps have a full functional phage life cycle, albeit the P1-like plasmids do presumably propagate as plasmids (otherwise they would have been lost) and all these elements are effectively on genomes and available for recombination. The conversion of P-Ps to integrative prophages may have resulted from P-Ps that survived because they were able to integrate the chromosome (e.g. using the Cre-Lox recombination system[63] or by the action of a transposable element). Some defense systems are known to target circular MGEs[64,65] and integration in the chromosome may allow the element to escape them. Also, integrated P-Ps may replicate with the bacterial

chromosome and no longer require plasmid replicases and segregation systems. Integrating the chromosome may lower the cost of these elements and prevent problems of compatibility between replicons. On the other hand, integration in the chromosome may inactivate genes and implicates that the multi-copy state is lost. Further work will be needed to assess the costs and benefits associated with the transitions between P-Ps and these novel P1-like integrative prophages.

The observed conversion of a P-P, an element transferring between cells as a phage (in capsids), to a mobilizable plasmid that moves between cells by conjugation, is a fascinating phenomenon from the perspective of the MGE because it represents a complete change of the mobility type. This conversion results from the loss of phage genes and acquisition of *oriTs* and other genetic elements implicated in conjugation. These gains may compensate the loss of the ability to propagate in viral particles since they allow the elements to transfer horizontally by conjugation in trans (i.e., mobilization by an autonomously conjugative element). While the precise order of events remains to be demonstrated, we found many ARGs in these novel plasmids which suggests that the acquisition of these genes increase the size of the replicon that made the P-P incapable of packaging its entire genome. This inactive element was not lost from populations and instead diversified as a plasmid, possibly because it provides antibiotic resistance to its host. Subsequently, exchanges with other plasmids may have resulted in the acquisition of functions permitting the element to salvage its capacity for horizontal transmission. In the present cases, this has led to the acquisition of genetic elements allowing the mobilization by a conjugative element. In sum, not only phages and plasmids exchange genetic material from P-Ps, they sometimes have P-Ps as their common ancestor.

In conclusion, the analysis of gene flow between three types of MGEs showed that gene flow is highest between plasmids, intermediate between P-Ps, and less frequent between phages. This is just a relative scale, we found many cases of genetic exchanges between phages (as extensively published before[8,37,66]). Genetic exchanges between phages and plasmids seem to take place at low frequencies. In contrast, exchanges of these two types of elements with P-Ps are much more frequent. Hence, P-Ps may play a key role in driving gene flow between the other elements. It has been proposed that all the world's a phage, i.e. that phage genomes are mosaics accessing a large common genetic pool by gene exchanges[67]. Our results suggest that this view can be extended to the broader pool of mobile genetic elements.

These results are important because whether a gene is encoded in a phage or a plasmid can impact its expression and spread in the population. Genes in phages and P-Ps can be overexpressed during the lytic cycle, as described for the Shiga toxin, in which case they may provide stronger phenotypes than in plasmids. In contrast, genes in plasmids and P-Ps can be at a higher copy number than the chromosome (and that of the integrated prophages) during a long period of time, which provides opportunity to evolve faster[68]. Polyploidy of a P-P was recently shown to facilitate the rapid spread of mutated alleles in a population[69]. Life cycles and fitness costs of plasmids and phages are very different. While many of the former may impose some burden on the fitness, the latter will often lead to cell death. Hence, the transfer of a gene for an accessory trait between these different types of MGEs may be important for its spread by horizontal gene transfer across Bacteria and for its subsequence maintenance in genomes.

## Methods
### Genomes of bacteria, phages, plasmids and phage-plasmids
We retrieved all genomes, having plasmids or predicted prophages, from the non-redundant NCBI RefSeq database[70] (last accessed in March, 2021). This dataset has the fully assembled genomes of 16,985 bacterial strains which include 20274 plasmids, and in which we have predicted 50262 prophage regions by VirSorter2[71] v.2.2.3 (default parameters with the '−min-length 1500' option). Moreover, this

dataset also includes complete genomes of 3585 phages and 1416 P-Ps that were identified using a dedicated procedure (that was detailed before)[13,32].

### Gene homology and similarity between mobile genetic elements
The similarity between mobile genetic elements (MGEs) was assessed by the relatedness of their gene repertoires weighted by the protein sequence identity (wGRR), as described in previous work[13,37]. We first computed the bi-directional best hits (BBH) of all pairs of plasmids, P-Ps and (pro-) phages. We used MMseqs2[72] (v. 13-45111) with '-s 7.5 and -a' to make an all-vs-all protein comparison between all MGE genome pairs. BBHs were extracted using a customized R-script if the alignments met the following criteria: evalue $< 10^{-4}$, identity >35% and the alignment covering at least 50% of both gene sequences. The wGRR was calculated as:

$$wGRR(AB) = \frac{\sum_i^P id(A_i B_i)}{min(\#A, \#B)} \qquad (1)$$

$A_i$ and $B_i$ are the $i$th BBH out of $P$ total pairs between two MGE genomes (A,B). The number of genes of the smaller element is $min(\#A, \#B)$, and the sequence identity between the BBH pair is $id(A_i, B_i)$. The wGRR takes into account the number and identity of BBH between the elements. It will be very high when BBH are a large fraction of the smallest element and the genes/proteins are very similar. When either the fraction of BBH or sequence similarity are intermediate, the wGRR will have intermediate values between 0 and 1.

### Grouping of phages, plasmids and phage-plasmids
Phages, plasmids and P-Ps were clustered into related groups using state-of-the-art techniques for each type of element. For phages, we used vConTACT2[33] (v. 0.9.19, default parameters), which grouped 2412 of the 3585 phages into 258 viral clusters (VC). To cluster plasmids, we used COPLA[34] (default parameters, v. 1.0) which uses average nucleotide identity between plasmids to cluster them in plasmid taxonomic units (pTUs), and, in addition, performs Inc-typing by using PlasmidFinder[73]. We could assign 9383 of the 20274 plasmids to 355 pTUs. Finally, we grouped P-Ps by clustering the wGRR matrix that includes the comparisons between all pairs of elements using the Louvain clustering algorithm, following our previous work[13]. A total of 513 out of 1416 were grouped into 9 well-related groups (for which there is at least one element verified experimentally) and 615 into 22 broadly-related communities. The mean wGRR between groups was calculated by summing up all values between members of the two groups and dividing the sum by the number of pairs between the two groups.

### Gene exchanges between mobile genetic elements
To distinguish genetic exchanges from homologous sequences that arose from common ancestry upon vertical descent, we followed an approach previously published[37,74]. Briefly, we identified the pairs of proteins that were bi-directional best hits (BBH) between two elements with high sequence similarity (>80% identity covering >80% of both proteins). In parallel, we computed the wGRR between the two elements to have a measure of their global similarity. We then searched for pairs of elements with low relatedness (wGRR =<0.1) and encoding pairs of highly similar homologs (BBH) (table with assignments is available on figshare, https://doi.org/10.6084/m9.figshare.23618667). All BBH that fall into this category were classed as recombining genes (RGs).

Our preliminary analysis found large sets of RGs when comparing megaplasmids (>300 kb). Current literature suggests that megaplasmids have different evolutionary dynamics than the other plasmids. They often persist long times in lineages and can be chromids or secondary chromosomes[75]. This is a problem, because in such cases

one cannot exclude the possibility that phage functions in megaplasmids are caused by the integration of phages that have evolved to integrate secondary chromosomes or chromids. If a prophage integrates a secondary chromosome (labeled as plasmid), this could artifactually add RGs between plasmids and phages. To limit this problem, we rejected from our analysis pairs of elements where the number of highly-related BBH was higher than 25 (rejecting 53834 genes). Note that this is a rare event that only affects comparisons between very large replicons. In most cases if there are 25 highly similar genes, then wGRR≫0.1 (resulting in their exclusion from further analysis).

Genes that were not classed as RGs were considered as not recombining (NRG). Some genes lacked homologous sequences and were therefore referred as NRGs with no homologs (NRG-nh) (Fig. 2). Of note, NRG and NRG-nh may have been the result of gene exchanges with MGEs not present in our analysis. Hence, this is a conservative estimate of the number of exchanged genes.

## Quantification of genetic exchanges between mobile genetic elements

To assess quantitatively the gene flow between different phages, plasmids and P-Ps, we first clustered all RGs into families by sequence similarity. We used a strict threshold of sequence similarity (≥80% sequence identity and ≥80% coverage based on amino acid sequences) and then used a single-linkage algorithm (each gene in a family has a least one other highly similar member) to produce the clusters (Supplementary Dataset S4).

We then analyzed each RG family in terms of the elements showing evidence of genetic exchanges. For each family there are potentially six different types of exchanges: within a given type of MGE (phage < ->phage, P-P < -> P-P, plasmid < ->plasmid) and between types (plasmid < ->P-P, P-P < ->phage, phage < ->plasmid). We counted the types of exchange present in each family. For instance, if an RG family is in four plasmids and two P-Ps, we mark the presence of exchanges between plasmids, between P-Ps and between P-Ps and plasmids (Fig. S5B). The counts of presences of types of events is better than simply counting the number of RG elements because many similar MGEs may have the same RG that they recently inherited from an ancestor that exchanged genes with another element. Counting RG assignments would thus vastly inflate the numbers of ancestral events of exchange (Fig. S5BC).

## Functional annotation of genes

RGs and NRGs were categorized in terms of function by comparing their sequences to several specialized and curated databases. To annotate phage functions, we performed profile-to-proteins comparisons with HMMER[76] v. 3.3.2 using PHROGs[38]. Profiles of the category "Unknown" were excluded. If a gene was assigned to multiple categories, only the one with the best hit was kept (highest bitscore). Plasmid replication and partition functions were annotated using specific profiles[77]. In all the searches using HMM profiles, a positive hit was assigned if the alignment covered at least 50% of the profile with a domain i-Evalue < 10^{-3} (following the criteria of MacSyFinder v2[78]).

Genes involved in defense mechanisms were identified by using profiles from DefenseFinder[79] v. 1.0.9. We aimed to include all hits, i.e. also the incomplete systems (orphan genes), since MGEs often have compact versions of defense systems[41]. We followed two approaches, one time using the "−cut_ga" (as done in DefenseFinder[79]) and another time by applying a threshold of 50% profile coverage and a domain i-Evalue < 10^{-3} (referred as Defense-05). This was done to minimize false-positives since we allowed for the identification of systems that may not be complete. This second set has 51.6% fewer hits (Supplementary Dataset S5). Antibiotic resistance genes were detected using AMRFinderPlus[80] v. 3.10.18 with default parameters.

To detect different types of recombinases and transposases we used a similar approach as described in previous work[37]. Briefly, for

recombinases, we used the profiles[37] with the indicated thresholds on the bitscore (for Sak4 >=28 and gp2.5 > = 20) and the "--cut_ga" option for the PFAMs. For transposases, we used profiles of ISEScan[81] and, in addition, the protein sequence database of ISFinder[82]. With the ISEScan profiles, we annotated genes as transposases if they covered at least 60% of the sequence with a sequence evalue < 10^{-5}. For the comparison of protein sequences of ISfinder we used MMseqs2 and positive hits were assigned if they had at least 80% protein sequence identity to the reference transposases and if alignment is covering at least 80% of the reference sequence. Some genes were annotated by ISEscan and by profiles of recombinases or with sequences of transposases from ISfinder (but the latter did not overlap). For clarity, we classed the recombinases as recombinases and the transposases as ISFinder transposases (since ISEscan profiles are based on ISFinder sequences). Of note, these categories of recombinases and transposases strongly overlap with the PHROG category "integration and excision". In particular, 97.5% of plasmid-encoded recombinases and transposases match also this PHROG category, 95.8% in P-Ps and 48.9% in phages (Supplementary Dataset S3).

Virulence factors were identified using a specialized database (VFDB[83], last accessed on 17.12.2022), which includes experimentally verified and predicted proteins. The protein sequences of the database were retrieved and compared to all MGE genes using MMseqs2. A protein was classed as a virulence factor if the alignment covered at least 80% of the virulence gene from VFDB with sequence identity >80%.

## Statistical comparison of recombining and non-recombining genes

Annotated genes were grouped into functional categories and into groups of recombining and non-recombining genes (RGs and NRGs) (Supplementary Dataset S5). In addition, all RGs were also grouped into genes exchanging between different types (bMGE) and within the same type of MGEs (wMGE) (Supplementary Dataset S6). Over- and underrepresentation of annotated genes in the different categorical groups were tested in contingency tables using the exact Fisher's test with the Benjamini-Hochberg multiple-testing correction. To assess the magnitude of differences per functional category, we computed difference-sum ratios ($R_{Diff-Sum}$) as follows:

$$R_{Diff-Sum} = \frac{O - E_{H0}}{O + E_{H0}} \qquad (2)$$

With the observed fraction (O) being the number of annotated genes in each group of RGs, NRGs, bMGEs and wMGEs, and $E_{H0}$ being the expected value under the tested null hypothesis (equal distribution within the groups). $R_{Diff-Sum}$ ranges between -1 and +1, with positive values indicating an over- and negative values an underrepresentation. $R_{Diff-Sum}$ values close to 0 indicate a nearly equal distribution (no over-/underrepresentation), and closer to 1 (or -1) show strong differences. To give an example (numbers taken from Supplementary Dataset S5), 3.01% of all genes in P-Ps (RGs and NRGs) are annotated as "Integration and Excision" by using PHROG profiles, meaning that $E_{H0}$ is equal to 3.01%. After allocating all genes into RGs and NRGs, we observe that 8.60% of the RGs and 1.87% of the NRGs are in the category of "Integration and Excision" ($O_{RG} = 8.60\%$ and $O_{NRG} = 1.87\%$), resulting in an $R_{Diff-Sum} = +0.48$ for RGs and an $R_{Diff-Sum} = -0.23$ for NRGs.

## Identification of P1-related plasmids and integrative prophages

To identify integrative prophages and plasmids that are related to P1-like P-Ps, all prophage regions and plasmids were searched for elements with a wGRR >= 0.1 to P-Ps from the P1 group (only pairs of elements having at least 10 homologs). This procedure identified 12 prophage regions and 45 plasmids. These elements were visually inspected, and subsequently pooled for further analysis. Their genes

were compared to gene families (persistent and shell) conserved within the complete P1 group (based on 123 P1g1 and 26 P1g2 P-P genomes)[32]. They were assigned to a gene family when they met the criteria of at least 80% identity in an alignment covering 80% of the lengths of both genes.

## Phylogenetic tree of P1-like mobile genetic elements

P1-like MGEs were compared to produce a phylogeny. First, we selected closely related MGEs, i.e. those having at least 75% of conserved P1 gene families that occur in at least 75% of the genomes. This resulted in a subset of 119 P-Ps of the P1 subgroup 1 (P1g1), 3 putative integrative prophage regions and 38 plasmids. The pangenome of all these elements was computed using PanACoTA[84] v.1.2. Highly similar homologs were searched using MMseqs2[72] and kept when protein sequence identity was more than 80% and the alignment covered at least 80% of their lengths. The gene families were then built from these pairs by single-linkage clustering. We selected the persistent gene families from the pangenome, i.e. we picked the gene families occurring in a single copy in at least 90% of the P1-like MGE genomes (strict persistent genome with "-t 0.9"). The nucleotide sequences of the 17 gene families of the persistent genome were aligned separately with MAFFT[85] "−auto" using PanACoTA v.1.2 (PanACoTA aligns the corresponding proteins and then back-translates them to DNA to obtain a more accurate alignment). The multiple alignments were then concatenated and used as input to IQ-TREE 2[86] v. 2.0.6 to compute a phylogenetic tree. Tree robustness was assessed by computing 1000 ultrafast bootstraps[87] and the best model was searched with ModelFinder[88] using "-m TEST -B 1000" as parameters. The best substitution model was $SYM + I + G4$, a symmetric model with equal base frequencies but unequal rates. We rooted the tree in two ways: using a mid-point root and using an outgroup. We re-ran PanACoTA with the same parameters but included two genomes of the P1g2 group (phage D6 (NC_050154) and NZ_CP066033[13]). Because P1g2 elements are more divergent, the strict persistent genome was of only 15 gene families in this case (instead of 17). Phage D6 was set as the basal clade. In the outgroup-rooted tree as well as in and the midpoint-rooted one (without members of the P1g2 group) most of the plasmids were placed in a common branch (Fig. S14). Since the outgroup-rooted tree had fewer branches with high bootstrap support, caused by the use of a smaller multiple alignment, we used the midpoint-root version.

## Reporting summary

Further information on research design is available in the Nature Portfolio Reporting Summary linked to this article.

## Data availability

The genomics data employed in this study is openly accessible and can be obtained from the NCBI database (https://www.ncbi.nlm.nih.gov/) using the respective gene, protein, or genome IDs. The assignments of recombining genes generated in this study have been deposited in figshare https://doi.org/10.6084/m9.figshare.23618667.

## Code availability

Custom R scripts (wGRR, gene clustering by single linkage, quantification of gene flow, enrichment tests) developed for this study are available in a figshare repository https://doi.org/10.6084/m9.figshare.23618667.

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

## Acknowledgements

Equipe FRM (Fondation pour la Recherche Médicale): EQU201903007835, Laboratoire d'Excellence IBEID Integrative Biology of Emerging Infectious Diseases [ANR-10-LABX-62-IBEID]. We extend our special thanks to Jorge Moura de Sousa for engaging and enlightening discussions about gene flow between phages and other mobile genetic elements. Additionally, we express our gratitude to Manuel Ares-Arroyo, Charles Coluzzi, and Julien Guglielmini for providing valuable insights into the mobility of phage-related plasmids and for their thoughtful comments on the interpretation of the phylogenetic data. This work used the computational and storage services (TARS & MAESTRO cluster) provided by the IT department at Institut Pasteur, Paris.

## Author contributions

EP and EPCR conceptualized the research study. EP performed the data analysis visualization and drafted the initial manuscript. EPCR provided critical revisions to the manuscript and gave valuable interpretations to the results. EP and EPCR edited and refined the manuscript.

## Competing interests

The authors declare no competing interests.
