## [Peer Review File · Nature Communications]

Phage-plasmids promote recombination and emergence of phages and plasmidsReviewer #1 (Remarks to the Author):

The manuscript by Pfeifer and Rocha contains an extensive analysis of recent gene exchange events between phages (Ps), plasmids (Ps), and what the authors call "phage-plasmids" (P-Ps). The analysis is based on the comparison of the numbers and functional categories of genes that could be found as common to all three or either two types of these mobile genetic elements (MGEs), as identified by classifying the genes of all the analyzed elements by high sequence similarity (in the most cases 80% coverage and 80% identity) and verification whether the particular functional categories of similar genes are present only in one or in more types of elements. Based on the obtained results the authors divided the analyzed genes into three categories: recombining (RGs), non-recombining (NRGs), and non-recombining with no homologs (NRG-nh) if the gene was unique (the latter was not taken into consideration in the calculation of exchange events). This in turn enabled them to calculate roughly which functional categories of genes undergo the exchange between different classes of the tested MGEs, to compare the frequency of the exchanges within and between the tested groups of MGEs, as well as to determine the functional types of genes that are preferentially exchanged between particular types of MGEs. The calculations clearly show a key role of P-Ps as intermediates in the exchanges between plasmids and phages, despite the relatively low number of P-Ps elements as compared to plasmids or phages. The methods of data preparation as well as calculations are sufficiently described. The conclusions are based on the analysis of the extensive data set and are convincing. Additionally, they provide the first example of such extensive analysis of this kind. Although the manuscript is interesting and provides a novel view of the directions and preferences of gene flows between the analyzed MGEs, the text is not easy to read, contains some ambiguities, and could be improved.

The name phage-plasmid which is used by the authors after Ravin et al. (1999) is not very fortunate. The original name that has been commonly used for phages that lysogenize as plasmids is "plasmid-prophage" or "prophage-plasmid"(see e.g., Schwesinger and Novick, 1975, DOI: 10.1128/jb.123.2.724-738.1975; Tucker and Pemberton, 1978, DOI: 10.1128/jb.135.1.207-214.1978; Sternberg and Austin, 1981, DOI: 10.1016/0147-619x(81)90075-5; and many other publications). Phage DNA stably existing in a cell is a prophage, not a phage. It has to have its lytic genes repressed and is just DNA. The name "phage-plasmid" is confusing, as the name "phage" applies to a whole phage virion with DNA inside, not just DNA alone. This should be mentioned, to avoid possible confusion, or, better, even corrected in the manuscript, to avoid further propagation.

The introduction section would benefit from a clear definition of what is considered by the authors to be P-Ps. Is this name strictly reserved for functional plasmid-prophages that retain the ability to enter and complete lytic development or is it extended to MGEs that contain plasmid and phage genes (including the phage lysis-lysogeny control module) but not necessarily retain the ability to enter and complete lytic development, e.g., due to the loss or certain essential phage genes or due to the acquisition of too many genes to be packed as a whole to a relevant phage capsid? Although the authors mention in the discussion section that their analysis cannot ascertain if the analyzed P-Ps have a full functional phage life cycle this information should be provided in the introduction section to avoid confusing readers. Additionally, it would be helpful to specify in the introduction that while defective plasmid-prophages that cannot enter and complete lytic development cannot transfer their genetic material between cells they are still accessible for the recombination with plasmid and with phages, which justifies their inclusion in the analysis.

The discussion section would be easier to read if it contained a list of the most important conclusions resulting from the performed analysis. Additionally, it could be simplified, for better clarity.

More specific comments are below.

L. 25: Replace "became mobilizable" with "become plasmids mobilizable" to avoid confusion of what is plasmid and what is P-Ps

L. 488-491: The size of DNA fragments that can be incorporated into an integrated prophage or

into a plasmid-prophage by recombination without the loss of its ability to enter and complete lytic development, is limited by the size of the capsid. Thus, the tolerance to the large extra fragments of genetic material correlates with the size of redundant genome fragments in virion DNA. This should be better emphasized by the authors.

L. 534: Replace "phages" with "prophages" to avoid confusion.

L. 538, and elsewhere in the text: Plasmid-prophages are the prophages that lysogenize as plasmids. Thus, to avoid confusion the authors might replace the name "prophages" with "integrated prophages" wherever they mean the integrated prophages.

Reviewer #2 (Remarks to the Author):

In this article, the authors study recent gene exchanges in different mobile genetic elements, i.e. phages, plasmids, and phage-plasmids (PPs), that designate temperate phages that are maintained in an episomal state as prophages. To the best of my knowledge, gene flows between different types of MGE had so far been only poorly addressed. The authors found more genes exchanged among plasmids than among phages, with PP having an intermediate level. However, the question of the bias that could result from the very different sizes of the different MGE groups (3585 phages, 20274 plasmids and 1416 PP) is not quantitatively addressed, which prevents a robust conclusion (indeed, intuitively it seems that the larger a group of genomes, the higher the probability than a gene in a genome has a homolog in another genome). In addition, the comparison is difficult as MGE groups (phage VC and plasmid pTU) are not constructed with the same tools, so the diversity within groups might be different, and hence the meaning of recombination events counted between and within groups is difficult to apprehend (and it seems from Figure 1 that phages are overall more related to each other than plasmids).

In a second step, the authors grouped recombining genes (RG), to quantify the exchanges between MGE types. By counting exchanges between MGE types at the family level, they found more events between PPs and plasmids or phages than between phages and plasmids. Yet, one wonders whether this could result from a bias of the definition of PPs (ie plasmids that possess phage genes). Would it be possible to disentangle recent recombination events from ancient ones that gave birth to PPs (whatever the direction of the evolution).

Overall, the authors nevertheless demonstrated unambiguously genetic exchanges between phages and plasmids, and they also show a nice example of relatedness between P1-like PP and integrated prophages. However, the claim that PPs drive gene flows between phages and plasmids, if suggested, is not demonstrated. The results only show that PPs recently exchanged genes with both phages and plasmid, without demonstrating a role for PPs in the passage of a gene from a plasmid to a phage. The gene exchanges observed between phages and plasmids could also have occurred directly. I would therefore recommend lowering the conclusion in throughout the text.

-Minor points:

Introduction: indicate very early that "phages" do not include PPs (before lines 98-99)

L36-37: It is known from a very long time that some prophages are not integrated in the bacterial genome and replicate as plasmids

Paragraph L102-125: indicate that this result is expected since PPs have by construction genes of phages and genes of plasmid

L169-170: indicate in the text that Fig S4 is a sketch of the methodology used

Fig S4 legend L219 : "If a PP has an RG in a plasmid", shouldn't it be "If a PP share a RG with a plasmid"? Other inconsistencies in the legend

L264 : "virulence factor"

Reviewer #3 (Remarks to the Author):

In this study by Pfeifer and Rocha, the authors investigate the gene flow between three types of MGEs: phage-plasmids (P-Ps), phages and plasmids, focusing on revealing where P-Ps stand in these gene exchange events. To do this, they searched for genes exchanged between divergent MGEs, characterised their functions, determined the frequency of the exchange events, and explored their impact on generating new MGEs. The authors report that exchanged genes are far more frequent between P-Ps and plasmids or phages than between the latter two, that exchanged genes can encode functions important for the host and MGE adaptation and that the dynamics of gene exchange may drive the emergence of plasmids and prophages from P-Ps.

This study builds on previous work by the authors on identifying P-Ps and characterising their diversity and contribution to the carriage of adaptive traits such as AMR genes. Here, the authors provide a thorough view of gene flow between MGEs, a comprehensive framework for the systematic measurement of gene exchange, and make P-Ps central stage, which I consider novel and significant to the field of horizontal gene transfer. I find the notion of P-Ps giving birth to plasmids via the acquisition of mobility determinants particularly fascinating, and I see researchers revisiting this study to find whether their gene of interest is subject to frequent horizontal transfer. I only have a couple of major comments and a number of minor comments that the authors may want to consider.

Major comments:

P-Ps contribution to gene exchange between phages and plasmids. Across the manuscript (e.g. L19-21, L134), the authors conclude that gene exchanges between phages and plasmids involve P-Ps. Their results show that based on gene repertoire relatedness, P-Ps sit in the middle between plasmids and phages and that P-Ps have exchanged many more genes with phages and plasmids than these two MGE types between each other. Indeed, these findings are valuable and suggest that P-Ps play a role in bridging transfer between phages and plasmids (as the authors state in L183-184), but they are not absolute proof that this is the case. Such a conclusion would require demonstrating that a given gene passed through P-Ps before reaching phages or plasmids, a tripartite sharing (which the authors report were detected in few numbers - L181). Under the framework established in this study, however, it would be impossible to rule out that the gene did not move directly from phage to plasmid, given the high sequence similarity the three genes would share. I recommend softening or adjusting all relevant statements in the manuscript (including in the abstract and results headings) to reflect this distinction between what the results show and suggest.

Number of MGEs analysed. There are discrepancies between the number of phages and plasmids described in the results and methods: 3585 (L104, L619) VS 3725 (L595) phages and 20274 (L105, L622) VS 21550 (L593) plasmids. This reviewer could not find an explanation for these differences in the text. Please ensure that all numbers check or clarify why they do not.

L653-655. These sentences (the couple of sentences at the end of the "Gene exchanges between mobile genetic elements" methods section) are very important. Please integrate them into the discussion.

Code availability. I urge the authors to save readers' time in emailing them to request the code and their own time responding to those emails by making the scripts available in a public repository.

Minor comments:

L109. "than have a ...". Perhaps you meant "that have a"?

L178-180. To what extent do plasmid and phage functions in P-Ps explain this?

L283 (Figure 3 Legend). "Blue underrepresentation". Do you mean black? I can't find blue stars.

Figure S9. The similarity in the bottom (Defense) pairwise comparison seems low, below the threshold established to detect RGs. Is this correct?

Ancestral and conserved genes. How do genes under selection to maintain sequence conservation fit in the RGs framework established in this study? These would share high sequence similarity but were not exchanged recently. Would your approach identify these as RGs? Can the authors comment on this?

Synteny plot. Pairwise comparison maps are recurrent in the paper, mostly featured in supplementary figures; however, there is no description in the methods of the software used to make these maps. Please add this information.

Figure S12. The colour code described in the legend for prophages and plasmids needs to be corrected.

L427-429. Please clarify in the text whether the three prophages have lost the plasmid functions.

Function categories among RGs. It would be very informative to see what proportion of the RGs were annotated as part of the main functional categories. Can you add a plot in Figure S5 breaking down the proportion of RGs in P-Ps, phages and plasmids annotated as Phage function, Plasmid function, AMR, Virulence, Defense and Transposases/Recombinases?

L606. Was the wGRR calculated in R or Python? Please make the code available. This will facilitate other researchers implementing your approach.

L647-649. Indicate how many elements were rejected.

Single-linkage algorithm. How was this implemented? Was it done in R or Python as part of an existing package? Please provide the relevant details (e.g. function and package used).

RESPONSE TO REVIEWER COMMENTS

Reviewer #1 (Remarks to the Author):

The manuscript by Pfeifer and Rocha contains an extensive analysis of recent gene exchange events between phages (Ps), plasmids (Ps), and what the authors call "phage-plasmids" (P-Ps). The analysis is based on the comparison of the numbers and functional categories of genes that could be found as common to all three or either two types of these mobile genetic elements (MGEs), as identified by classifying the genes of all the analyzed elements by high sequence similarity (in the most cases 80% coverage and 80% identity) and verification whether the particular functional categories of similar genes are present only in one or in more types of elements. Based on the obtained results the authors divided the analyzed genes into three categories: recombining (RGs), non-recombining (NRGs), and non-recombining with no homologs (NRG-nh) if the gene was unique (the latter was not taken into consideration in the calculation of exchange events). This in turn enabled them to calculate roughly which functional categories of genes undergo the exchange between different classes of the tested MGEs, to compare the frequency of the exchanges within and between the tested groups of MGEs, as well as to determine the functional types of genes that are preferentially exchanged between particular types of MGEs. The calculations clearly show a key role of P-Ps as intermediates in the exchanges between plasmids and phages, despite the relatively low number of P-Ps elements as compared to plasmids or phages. The methods of data preparation as well as calculations are sufficiently described. The conclusions are based on the analysis of the extensive data set and are convincing. Additionally, they provide the first example of such extensive analysis of this kind. Although the manuscript is interesting and provides a novel view of the directions and preferences of gene flows between the analyzed MGEs, the text is not easy to read, contains some ambiguities, and could be improved.

We thank the reviewer for accepting the task and for the positive assessment of our work. We have read and changed the manuscript in order to make it clearer.

1.1 The name phage-plasmid which is used by the authors after Ravin et al. (1999) is not very fortunate. The original name that has been commonly used for phages that lysogenize as plasmids is "plasmid-prophage" or "prophage-plasmid"(see e.g., Schwesinger and Novick, 1975, DOI: 10.1128/jb.123.2.724-738.1975; Tucker and Pemberton, 1978, DOI: 10.1128/jb.135.1.207-214.1978; Sternberg and Austin, 1981, DOI: 10.1016/0147-619x(81)90075-5; and many other publications). Phage DNA stably existing in a cell is a prophage, not a phage. It has to have its lytic genes repressed and is just DNA. The name "phage-plasmid" is confusing, as the name "phage" applies to a whole phage virion with DNA inside, not just DNA alone. This should be mentioned, to avoid possible confusion, or, better, even corrected in the manuscript, to avoid further propagation.

We are thankful for this suggestion and the references. When we decided on a name, we used the term "phage", following Ravin and some other authors to emphasize that phage-plasmids are a type of temperate phage, i.e. that they are packaged in viral particles and to exclude cases where the prophage is described as co-integrated in a plasmid as in Schwesinger and Novick, 1975. We agree with the reviewer that calling them prophages is also relevant. Since this is the fourth paper on a series of studies we have done on these elements, we prefer to maintain consistency in the naming (of note, the acronym we use P-P actually works for both). Yet, we now explicitly make the point raised by the reviewer.

1.2 The introduction section would benefit from a clear definition of what is considered by the authors to be P-Ps. Is this name strictly reserved for functional plasmid-prophages that retain the

ability to enter and complete lytic development or is it extended to MGEs that contain plasmid and phage genes (including the phage lysis-lysogeny control module) but not necessarily retain the ability to enter and complete lytic development, e.g., due to the loss of certain essential phage genes or due to the acquisition of too many genes to be packed as a whole to a relevant phage capsid? Although the authors mention in the discussion section that their analysis cannot ascertain if the analyzed P-Ps have a full functional phage life cycle this information should be provided in the introduction section to avoid confusing readers.

The method to identify P-Ps was described before. It searches for a minimal set of functions required for a plasmid to be a phage (packaging, lysis, tail, head, etc). Hence, these are not just plasmids with a few phage genes. This being a computational study we cannot exclude the possibility that some P-Ps are not functional anymore. We have explained this in the beginning of the results section.

1.3 Additionally, it would be helpful to specify in the introduction that while defective plasmid-prophages that cannot enter and complete lytic development cannot transfer their genetic material between cells they are still accessible for the recombination with plasmid and with phages, which justifies their inclusion in the analysis.

Thanks for pointing this out. This was added right after the sentence tackling comment 1.2.

1.4 The discussion section would be easier to read if it contained a list of the most important conclusions resulting from the performed analysis. Additionally, it could be simplified, for better clarity.

We have done this (while doing our best to respect space restrictions).

More specific comments are below.

1.5 L. 25: Replace "became mobilizable" with "become plasmids mobilizable" to avoid confusion of what is plasmid and what is P-Ps

This was re-written for clarity.

1.6 L. 488-491: The size of DNA fragments that can be incorporated into an integrated prophage or into a plasmid-prophage by recombination without the loss of its ability to enter and complete lytic development, is limited by the size of the capsid. Thus, the tolerance to the large extra fragments of genetic material correlates with the size of redundant genome fragments in virion DNA. This should be better emphasized by the authors.

We have explained this problem in more detail.

1.7 L. 534: Replace "phages" with "prophages" to avoid confusion.

This was done.

1.8 L. 538, and elsewhere in the text: Plasmid-prophages are the prophages that lysogenize as plasmids. Thus, to avoid confusion the authors might replace the name "prophages" with "integrated prophages" wherever they mean the integrated prophages.

Yes, we agree that prophages include phage-plasmids and integrative prophages. For better clarification, we added the distinction across the text.

Reviewer #2 (Remarks to the Author):

We appreciate the reviewer for accepting the task and for the positive evaluation of our work.

2.1 In this article, the authors study recent gene exchanges in different mobile genetic elements, i.e. phages, plasmids, and phage-plasmids (PPs), that designate temperate phages that are maintained in an episomal state as prophages. To the best of my knowledge, gene flows between different types of MGE had so far been only poorly addressed. The authors found more genes exchanged among plasmids than among phages, with PP having an intermediate level. However, the question of the bias that could result from the very different sizes of the different MGE groups (3585 phages, 20274 plasmids and 1416 PP) is not quantitatively addressed, which prevents a robust conclusion (indeed, intuitively it seems that the larger a group of genomes, the higher the probability than a gene in a genome has a homolog in another genome).

There is indeed an imbalance between the three datasets since there are many plasmids, fewer P-P and an intermediate number of phages. Yet, one should note that we search for highly similar genes in very distinct genomes and we only count once an exchange event between two groups of elements (exactly to avoid over-counting events in groups that are more abundant). We had considered to normalize this by the number of elements. We decided not to because an effective normalization should account for many factors that we find hard to control for (sampling, element size, gene number, etc). Importantly, if this was random, the number of exchanges would be lowest in the smallest sample (here P-Ps) and the highest between the two largest groups (phages and plasmids). Following the reviewer's request, we tested and confirmed this by conducting simulations in which we randomly labelled all MGEs (by assigning randomly labels of "phage", "plasmid" and "P-P" in the same proportion). We found more exchanges (than expected) between P-Ps and plasmids or phages and fewer between plasmids and phages. This further confirms our conclusions. We added this information in the text.

2.2 In addition, the comparison is difficult as MGE groups (phage VC and plasmid pTU) are not constructed with the same tools, so the diversity within groups might be different, and hence the meaning of recombination events counted between and within groups is difficult to apprehend (and it seems from Figure 1 that phages are overall more related to each other than plasmids).

The reviewer is worried that the grouping of elements leads to biases in the identification of the gene exchanges because different criteria are used for different elements. This was not well explained enough in our text. The grouping of the elements is only used in the analysis leading to the large graph in Figure 1, which is used just to display relations of homology but not to make any kind of subsequent quantification. In this graph analysis, we decided to use the standard way of grouping elements in each domain because we think this allows readers to recover more easily the groups they are familiar with. However, the analysis of gene exchange does not use the information of these groupings at all. The analysis is done across all elements. Hence, there is no bias associated with different grouping methods. This is now explained more in detail in the text.

2.3 In a second step, the authors grouped recombining genes (RG), to quantify the exchanges between MGE types. By counting exchanges between MGE types at the family level, they found more events between PPs and plasmids or phages than between phages and plasmids. Yet, one wonder whether this could result from a bias of the definition of PPs (ie plasmids that possess phage genes). Would it be possible to disentangle recent recombination events from ancient ones that gave birth to PPs (whatever the direction of the evolution).

The percent identity of the RG is very high relative to the percent identity of genes within families of P-P. To emphasize this, we added a histogram in the supplementary material showing frequency and sequence homology between gene pairs of recombining genomes (see Figure. S1). As a result, we only look at recent events, i.e. exchanges posterior to the formation of P-Ps. Also, note that cases of comparisons between plasmids (or phages) and P-Ps that might have arisen recently from co-integration of these phages and plasmids are automatically discarded by our method. This is because there are two conditions for a RG: high percent identity of the genes (see above) and very low wGRR of the elements. If a co-integration of a plasmid and a phage occurred recently then the wGRR with related plasmids and phages will be high and they will be discarded. This is now mentioned in the text.

2.4 Overall, the authors nevertheless demonstrated unambiguously genetic exchanges between phages and plasmids, and they also show a nice example of relatedness between P1-like PP and integrated prophages. However, the claim that PPs drive gene flows between phages and plasmids, if suggested, is not demonstrated. The results only show that PPs recently exchanged genes with both phages and plasmid, without demonstrating a role for PPs in the passage of a gene from a plasmid to a phage. The gene exchanges observed between phages and plasmids could also have occurred directly. I would therefore recommend lowering the conclusion in throughout the text.

Thank you for this comment. The frequent exchanges between P-P and the other two types of elements facilitates exchanges across plasmids and phages and we do show a few examples of exchanges across the three. Yet, these are not numerous (because we are extremely conservative, sample a small fraction of all elements, and here we require for a tripartite exchange which is certainly less frequent). We have therefore toned down our claim.

-Minor points:

2.5. Introduction: indicate very early that “phages” do not include PPs (before lines 98-99)

Following this comment and that of reviewer #1 we have extensively added integrative phages when that was the case. See changes made for #1.7 and #1.8.

2.6. L36-37: It is known from a very long time that some prophages are not integrated in the bacterial genome and replicate as plasmids

We have added a word to convey that they are mostly regarded as integrative. Note that a couple of lines below we add references to these early works.

2.7. Paragraph L102-125: indicate that this result is expected since PPs have by construction genes of phages and genes of plasmid

This is now mentioned.

2.8. L169-170: indicate in the text that Fig S4 is a sketch of the methodology used

Done.

2.9. Fig S4 legend L219 : “If a PP has an RG in a plasmid”, shouldn’t it be “If a PP share a RG with a plasmid”? Other inconsistencies in the legend

This is now corrected.

2.10. L264 : “virulence factor”

Done.

Reviewer #3 (Remarks to the Author):

In this study by Pfeifer and Rocha, the authors investigate the gene flow between three types of MGEs: phage-plasmids (P-Ps), phages and plasmids, focusing on revealing where P-Ps stand in these gene exchange events. To do this, they searched for genes exchanged between divergent MGEs, characterised their functions, determined the frequency of the exchange events, and explored their impact on generating new MGEs. The authors report that exchanged genes are far more frequent between P-Ps and plasmids or phages than between the latter two, that exchanged genes can encode functions important for the host and MGE adaptation and that the dynamics of gene exchange may drive the emergence of plasmids and prophages from P-Ps.

This study builds on previous work by the authors on identifying P-Ps and characterising their diversity and contribution to the carriage of adaptive traits such as AMR genes. Here, the authors provide a thorough view of gene flow between MGEs, a comprehensive framework for the systematic measurement of gene exchange, and make P-Ps central stage, which I consider novel and significant to the field of horizontal gene transfer. I find the notion of P-Ps giving birth to plasmids via the acquisition of mobility determinants particularly fascinating, and I see researchers revisiting this study to find whether their gene of interest is subject to frequent horizontal transfer. I only have a couple of major comments and a number of minor comments that the authors may want to consider.

We express our gratitude to the reviewer for the thorough evaluation and valuable constructive feedback on our work.

3.1 Major comments: P-Ps contribution to gene exchange between phages and plasmids. Across the manuscript (e.g. L19-21, L134), the authors conclude that gene exchanges between phages and plasmids involve P-Ps. Their results show that based on gene repertoire relatedness, P-Ps sit in the middle between plasmids and phages and that P-Ps have exchanged many more genes with phages and plasmids than these two MGE types between each other. Indeed, these findings are valuable and suggest that P-Ps play a role in bridging transfer between phages and plasmids (as the authors state in L183-184), but they are not absolute proof that this is the case. Such a conclusion would require demonstrating that a given gene passed through P-Ps before reaching phages or plasmids, a tripartite sharing (which the authors report were detected in few numbers - L181). Under the framework established in this study, however, it would be impossible to rule out that the gene did not move directly from phage to plasmid, given the high sequence similarity the three genes would share. I recommend softening or adjusting all relevant statements in the manuscript (including in the abstract and results headings) to reflect this distinction between what the results show and suggest.

This comment is similar to the one of 2.4. We agree with the criticism, explained better our results, and toned down the claim.

3.2 Number of MGEs analysed. There are discrepancies between the number of phages and plasmids described in the results and methods: 3585 (L104, L619) VS 3725 (L595) phages and 20274 (L105, L622) VS 21550 (L593) plasmids. This reviewer could not find an explanation for these differences in the text. Please ensure that all numbers check or clarify why they do not.

This was confusing in the manuscript. The two larger numbers mentioned in the methods section included the P-Ps prior to their detection. We have now clarified this better.

3.3 L653-655. These sentences (the couple of sentences at the end of the “Gene exchanges between mobile genetic elements” methods section) are very important. Please integrate them into the discussion.

We agree. They are now mentioned in the largely re-written first paragraph of discussion.

3.4 Code availability. I urge the authors to save readers’ time in emailing them to request the code and their own time responding to those emails by making the scripts available in a public repository.

R scripts that we used to analyze the data (wGRR, gene clustering into families, quantification of gene flow, and enrichment tests of functions) are now accessible through a link provided in the code availability section. These scripts, when used together with the supplementary tables, allow for the reproduction of the dataset.

Furthermore, we encountered a compatibility issue with gene IDs between our database and those used by NCBI (named Locus Tags). In a very few cases (0.03%), the CDS sections in the GenBank records lack a LocusTag specification (even though it encodes a CDS). To address this issue, we have excluded these few genes from our analysis, generated new figures and tables with corrected numbers. Of note, this did not affect any findings or statements.

Minor comments:

3.6 L109. “than have a ...”. Perhaps you meant “that have a”?

Indeed. Corrected.

3.7 L178-180. To what extent do plasmid and phage functions in P-Ps explain this?

11.2% of the gene families exchanged between P-Ps and phages or plasmids encode functions related clearly to plasmids (replication, partition) and phages (tail, lysis, head, and packaging, connector). However, to further clarify (since this point is also raised in comment 2.7), it's important to note that the composition of the graph in Figure 1 is partly influenced by P-Ps having homologs in both phages and plasmids. Yet, the abundance of RGs in P-Ps is not necessarily caused by the presence of such homologs. If there were no exchanges, then P-Ps would have no RGs with both phages and plasmids. Yet, since there are any exchanges, they may be facilitated by the presence of these homologues. But this is a biological finding, not a consequence of the methodology. We have rephrased the discussion to make this clearer.

We added the abovementioned proportion to the text.

3.8 L283 (Figure 3 Legend). “Blue underrepresentation”. Do you mean black? I can't find blue stars.

Color was changed to blue and red stars (as indicated in the figures text).

3.9 Figure S9. The similarity in the bottom (Defense) pairwise comparison seems low, below the threshold established to detect RGs. Is this correct?

In this plot, we intended to highlight exchanges that are mediated by transposases/recombinases. The defense genes are indeed not similar enough to be RGs, but the neighbouring transposases are. However, we agree that this might be confusing and since the reconstruction of this recombination event is not as clear as in other cases (such as ARG), we removed this example for clarity.

3.10 Ancestral and conserved genes. How do genes under selection to maintain sequence conservation fit in the RGs framework established in this study? These would share high

sequence similarity but were not exchanged recently. Would your approach identify these as RGs? Can the authors comment on this?

We understand that there is a concern that if a gene is highly conserved it may spuriously be classed as RG. This is why we set up such a low value of wGRR to compare MGEs. The value we use is <0.1 . This means that on average the sequence similarity between homologous genes is very low. If a third of the genes have homologs between elements (already a very low value that suggests high divergence), this means that each pair of homologs only has an average identity of 30%. In these conditions to have one gene with more than 80% identity would be extremely unusual. This is supported by analysis of sequence similarity of homologs between recombining MGEs (Figure S1). Most genes between recombining pairs are either not related or RGs. As a case in point, the most conserved genes in phages, such as terminases or capsids are not among the most frequently exchanged (between MGEs); they are among the few exchanged less frequently than expected (as shown in Figure S6). In contrast, gene expression regulators, which are known to evolve fast, are among the most exchanged.

3.11 Synteny plot. Pairwise comparison maps are recurrent in the paper, mostly featured in supplementary figures; however, there is no description in the methods of the software used to make these maps. Please add this information.

Thank you for this comment. We used `gggenomes` (which is unpublished), and it is now referred to it.

3.12 Figure S12. The colour code described in the legend for prophages and plasmids needs to be corrected.

Thanks. It is corrected.

3.13 L427-429. Please clarify in the text whether the three prophages have lost the plasmid functions.

All of the three prophages have still the replication partition genes (P1-like) encoded. This is now clarified in the text.

3.14 Function categories among RGs. It would be very informative to see what proportion of the RGs were annotated as part of the main functional categories. Can you add a plot in Figure S5 breaking down the proportion of RGs in P-Ps, phages and plasmids annotated as Phage function, Plasmid function, AMR, Virulence, Defense and Transposases/Recombinases?

Yes. It is added now.

3.15 L606. Was the wGRR calculated in R or Python? Please make the code available. This will facilitate other researchers implementing your approach.

Provided in the code availability section.

3.16 L647-649. Indicate how many elements were rejected.

In fact, all elements were kept. However, $>50k$ genes were rejected as RGs and is not clarified in the Methods section.

3.17 Single-linkage algorithm. How was this implemented? Was it done in R or Python as part of an existing package? Please provide the relevant details (e.g. function and package used).

The single linkage was done by first generating a graph from the RG assignment table and taking then components that have at least a single connection. The script (including functions and packages) is provided in the code availability section.

Reviewer #2 (Remarks to the Author):

All the reviewers' questions or remarks have been correctly addressed by the authors in the new version of the manuscript, that have been clarified, and I have no additional comments.

Reviewer #3 (Remarks to the Author):

After reading the revised version of the manuscript by Pfeifer and Rocha, I can confirm the authors have addressed all my comments, and I consider the paper to be ready for publication.